# A LIKELIHOOD BASED APPROACH TO DISTRIBUTION REGRESSION USING CONDITIONAL DEEP GENERATIVE MODELS

## ABSTRACT

In this work, we explore the theoretical properties of conditional deep generative models under the statistical framework of distribution regression where the response variable lies in a high-dimensional ambient space but concentrates around a potentially lower-dimensional manifold. More specifically, we study the large-sample properties of a likelihood-based approach for estimating these models. Our results lead to the convergence rate of a sieve maximum likelihood estimator (MLE) for estimating the conditional distribution (and its devolved counterpart) of the response given predictors in the Hellinger (Wasserstein) metric. Our rates depend solely on the intrinsic dimension and smoothness of the true conditional distribution. These findings provide an explanation of why conditional deep generative models can circumvent the curse of dimensionality from the perspective of statistical foundations and demonstrate that they can learn a broader class of nearly singular conditional distributions. Our analysis also emphasizes the importance of introducing a small noise perturbation to the data when they are supported sufficiently close to a manifold. Finally, in our numerical studies, we demonstrate the effective implementation of the proposed approach using both synthetic and real-world datasets, which also provide complementary validation to our theoretical findings.

## 1 INTRODUCTION

Conditional distribution estimation provides a principled framework for characterizing the dependence relationship between a response variable $Y$ and predictors $X$, with the primary goal of estimating the distribution of $Y$ conditional on $X$ through learning the (conditional) data-generating process. Conditional distribution estimation allows one to regress the entire distribution of $Y$ on $X$, which provides much richer information than the traditional mean regression and plays a central role in various important areas ranging from causal inference (Pearl, 2009; Spirtes, 2010), graphical models (Jordan, 1999; Koller and Friedman, 2009), representation learning (Bengio et al., 2013), dimension reduction (Carreira-Perpiñán, 1997; Van Der Maaten et al., 2009), to model selection (Claeskens and Hjort, 2008; Ando, 2010). Their applications span across diverse domains such as forecasting (Gneiting and Katzfuss, 2014), biology (Krishnaswamy et al., 2014), energy (Jeon and Taylor, 2012), astronomy (Zhao et al., 2021), and industrial engineering (Simar and Wilson, 2015), among others.

There is a rich literature in statistics and machine learning on conditional distribution estimation including both frequentist and Bayesian methods (Hall and Yao, 2005; Norets and Pati, 2017). Traditional methods, however, suffer from the curse of dimensionality and often struggle to adapt to the intricacies of modern data types such as the ones with lower-dimensional manifold structures.

Recent methodologies that leverage deep generative models have demonstrated significant advancements in complex data generation. Instead of explicitly modeling the data distribution, these approaches implicitly estimate it through learning the corresponding data sampling scheme. Commonly, these implicit distribution estimation approaches can be broadly categorized into three types. The first one is likelihood-based with notable examples including Kingma and Welling (2013), Rezende et al. (2014), Burda et al. (2015), and Song et al. (2021) . The second approach, based

on adversarial learning, matches the empirical distribution of the data with a distribution estimator using an adversarial loss. Representative examples include Goodfellow et al. (2014), Arjovsky et al. (2017), and Mroueh et al. (2017), among others. The third approach, which is more recent, reduces the problem of distribution estimation to score estimation through certain time-discrete or continuous dynamical systems. The idea of score matching was first proposed in Hyvärinen and Dayan (2005) and Vincent (2011). More recently, score-based diffusion models have achieved state-of-the-art performance in many applications (Sohl-Dickstein et al., 2015; Nichol and Dhariwal, 2021; Song et al., 2020; Lipman et al., 2022).

On the theoretical front, recent works such as Liu et al. (2021), Chae et al. (2023), Altekrüger et al. (2023), Stanczuk et al. (2024), Pidstrigach (2022) , and Tang and Yang (2023) demonstrate that distribution estimation based on deep generative models can adapt to the intrinsic geometry of the data, with convergence rates dependent on the intrinsic dimension of the data, thus potentially circumventing the curse of dimensionality. Such advancement has naturally motivated us to employ and investigate *conditional deep generative model* for conditional distribution estimation. Specifically, we explore and study the theoretical properties of a new likelihood-based approach to conditional sampling using deep generative models for data potentially residing on a low-dimensional manifold corrupted by full-dimensional noise. More concretely, we consider the following *conditional distributional regression* problem:

$$Y|X = V|X + \varepsilon, \tag{1}$$

where $X$ serves as a predictor in $\mathbb{R}^{\mathfrak{p}}$, $V|X$ represents the (uncorrupted) underlying response supported on a manifold of dimension $\mathfrak{d} \leq D$, $Y|X$ represents the observed response, and $\varepsilon \sim \mathsf{N}(0, \sigma_*^2 I_D)$ denotes the noise residing in the ambient space $\mathbb{R}^D$. Our deep generative model focuses on the conditional distribution $V|X$ by using a (conditional) generator of the form $G_*(Z, X)$, where $G_*$ is a function of a random seed $Z$ and the covariate information $X$. This approach is termed 'conditional deep generative' because the conditional generator is modeled using deep neural networks (DNNs). Observe that, when $\mathfrak{d} < D$, the distribution of $G_*(Z, X)$ is supported on a lower-dimensional manifold, making it singular with respect to the Lebesgue measure in the $D$-dimensional ambient space. We study the statistical convergence rate of sieve MLEs in the conditional deep general model setup and investigate its dependence on the intrinsic dimension, structure properties of the model as well as the noise level of the data.

## 1.1 LIST OF CONTRIBUTIONS

We briefly summarise the main contributions made in this paper.

- To the best of our knowledge, our study is the first attempt to explore the likelihood-based approach for distributional regression using a conditional deep generative model, considering full-dimensional noise and the potential presence of singular underlying support. We provide a solid statistical foundation for the approach by proving the near-optimal convergence rates for this proposed estimator.
- We derive the convergence rates for the conditional density estimator of the corrupted data $Y$ with respect to the Hellinger distance and specialize the obtained rate for two popular deep neural network classes: the sparse and fully connected network classes. Furthermore, we characterize the Wasserstein convergence rates for the induced intrinsic conditional distribution estimator on the manifold (i.e., a deconvolution problem). Both rates turn out to depend only on the intrinsic dimension and smoothness of the true conditional distribution.
- Our analysis in Corollary 2 suggests the need to inject a small amount of noise into the data when they are sufficiently close to the manifold. Intuitively, this observation validates the underlying structural challenges in related manifold estimation problems with noisy data, as outlined by Genovese et al. (2012).
- We show that the class of learnable (conditional) distributions of our method is broad. It encompasses not only the smooth distributions class, but also extends to the general (nearly) singular distributions with manifold structures, with minimal assumptions.

## 1.2 OTHER RELEVANT LITERATURE

The problem of non-parametric conditional density estimation has been extensively explored in statistical literature. Hall and Yao (2005), Bott and Kohler (2017), and Bilodeau et al. (2023) directly

tackle this problem with smoothing and local polynomial-based methods. Fan and Yim (2004) and Efromovich (2007) explore suitably transformed regression problems to address this challenge. Other notable approaches include the nearest neighbor method (Izbicki et al., 2020; Bhattacharya and Gangopadhyay, 1990), basis function expansion (Sugiyama et al., 2010; Izbicki and Lee, 2016), tree-based boosting (Pospisil and Lee, 2018; Gao and Hastie, 2022), and Bayesian optimal transport flow Chemseddine et al. (2024) among others.

In the context of conditional generation, we highlight recent work by Zhou et al. (2022) and Liu et al. (2021). In Zhou et al. (2022), GANs were employed to investigate conditional density estimation. While this work offers a consistent estimator, it lacks statistical rates or convergence analysis, and its focus is on a low-dimensional setup. In Liu et al. (2021), conditional density estimation supported on a manifold using Wasserstein-GANs was examined. However, their setup does not account for smoothness across either covariates or responses, nor do they address how deep generative models specifically tackle the challenges of high-dimensionality. Moreover, their assumption that the data lies exactly on the manifold can be restrictive. Our study shares some commonalities with the work of Chae et al. (2023), as both investigate sieve maximum likelihood estimators (MLEs). However, the fundamental problems addressed and the methodologies employed differ significantly, and our work involves technical challenges that span multiple scales. While Chae et al. (2023) concentrates exclusively on unconditional distribution estimation, our theoretical analysis necessitates much more nuanced techniques due to the conditional nature of our setup. This shift is noteworthy because it demands a more refined analysis of entropy bounds, considering two potential sources of smoothness - across the regressor and the response variables. Furthermore, our setting accommodates the possibility of an infinite number of $x$ values, which gives rise to a dynamic manifold structure, further compounding the intricacy of the problem at hand.

## 2 CONDITIONAL DEEP GENERATIVE MODELS FOR DISTRIBUTION REGRESSION

We consider the following probabilistic conditional generative model, where for a given predictor value $x$, the response $Y$ is generated by

$$Y = G_*(Z, x) + \varepsilon, \quad x \in \mathcal{X} \subset \mathbb{R}^{\mathfrak{p}}. \tag{2}$$

Here, $G_*(\cdot, x) : \mathcal{Z} \to \mathcal{M}_x$ is the unknown generator function, $Z$ a latent variable with a known distribution $P_Z$ and support $\mathcal{Z} \subset \mathbb{R}^{\mathfrak{d}}$ independent of the predictor $X$. The existence of the generator $G_*$ directly follows from Noise Outsourcing Lemma 3. This lemma enables the transfer of randomness into the covariate and an orthogonal (independent) component through a generating function for any regression response. We denote $\mathcal{M} := \cup_{x \in \mathcal{X}} \mathcal{M}_x \subset \mathbb{R}^D$ as the support of the image of $G_*(\mathcal{Z}, \mathcal{X})$ such as a (union of) $d$-dimensional manifold. We model $G_*(\cdot, \cdot) : \mathcal{Z} \times \mathcal{X} \subset \mathbb{R}^{\mathfrak{d}} \times \mathbb{R}^{\mathfrak{p}} \to \mathcal{Y} \subset \mathbb{R}^D$ using a deep neural network, leading to a *conditional deep generative model* for (2).

In the next section, we present a more general result in terms of the entropy bound (variance) for the true function class of $G_*$ and the approximability (bias) of the search class. We then proceed to a simplified understanding in the context of conditional deep generative models in subsequent sections.

### 2.1 CONVERGENCE RATES OF THE SIEVE MLE

In light of equation (2), it is evident that the distribution of $Y|X = x$ results from the convolution of two distinct distributions: the pushforward of $Z$ through $G_*$ with $X = x$, and $\varepsilon$ following an independent $D$-dimensional normal distribution. The density corresponding to the true distribution $P_*(\cdot|X = x)$ can thus be expressed as:

$$p_*(y|x) = \int \phi_{\sigma_*}(y - G_*(z, x)) \, dP_Z,$$

where $\phi_{\sigma_*}$ is the density of $\mathsf{N}(0, \sigma_*^2 I_d)$. We define the class of conditional distributions $\mathcal{P}$ as

$$\mathcal{P} = \left\{ P_{g,\sigma} : g(\cdot, x) \in \mathcal{F}, \sigma \in [\sigma_{\min}, \sigma_{\max}] \right\}, \tag{3}$$

where $P_{g,\sigma}$ represents the distribution with density $p_{g,\sigma} = \int \phi_\sigma(y - g(z, x)) dP_Z$. In this notation, $P_* = P_{G_*,\sigma_*}$ and $p_* = p_{G_*,\sigma_*}$. The elements of $\mathcal{P}$ comprise two components: $g$ originating from

the underlying function class $\mathcal{F}$, and $\sigma$, which characterizes the noise component. This class enables us to obtain separate estimates for $G_*$ and $\sigma_*$, furnishing us with both the canonical estimator for the distribution of $Y|X = x$ and enhancing our comprehension of the singular distribution of $G_*(Z, x)$, supported on a low-dimensional manifold.

Given a data set $\{(X_i, Y_i)\}_{i=1}^n$, the log-likelihood function is defined as $\ell_n(g, \sigma) = n^{-1} \sum_{i=1}^n \log p_{g,\sigma}(Y_i|X_i)$. For a sequence $\eta_n \downarrow 0$ as $n \to \infty$, a *sieve* maximum likelihood estimator (MLE) ([Geman and Hwang, 1982]) is any estimator $(\widehat{g}, \widehat{\sigma}) \in \mathcal{F} \times [\sigma_{\min}, \sigma_{\max}]$ that satisfies

$$\ell_n(\widehat{g}, \widehat{\sigma}) \geq \sup_{\substack{\sigma \in [\sigma_{\min}, \sigma_{\max}] \\ g \in \mathcal{F}}} \ell_n(g, \sigma) - \eta_n. \tag{4}$$

Here $\widehat{g} \in \mathcal{F}$ and $\widehat{\sigma} \in [\sigma_{\min}, \sigma_{\max}]$ are the estimators, and $\eta_n$ represents the optimization error. The dependence of $\widehat{g}$ and $\widehat{\sigma}$ on $n$ illustrates the sieve's role in approximating the true distribution when optimization is performed over the class $\mathcal{P}$. The estimated density $\widehat{p} = p_{\widehat{g}, \widehat{\sigma}}$ provides an estimator for $p_*(\cdot|\cdot)$, and $Q_{\widehat{g}}(\cdot|X = x)$ serve as the estimator for $Q_*(\cdot|X = x)$.

In this section, we formulate the main results, which provide convergence rates in the Hellinger distance for our sieve MLE estimator. The convergence rate was derived for any search functional class $\mathcal{F}$, with a brief emphasis on their entropy and approximation capabilities.

**Assumption 1** (True distribution). *Denote $\mu_X^*(x)$ as the distribution of $X$. We denote the true conditional densities as $p_* = \{p_*(\cdot|x), x \in \mathbb{R}^{\mathfrak{p}}\}$. It is natural to assume that the data is generated from $p_*$ from model (2) with some true generator $G_*$ and $\sigma_*$. We denote $Q_*(\cdot|X = x)$ (or $Q_{G_*}$) as the distribution of $G_*(Z, x)$ for some distribution $P_Z$.*

A function $g$ is said to have a composite structure ([Schmidt-Hieber, 2020]; [Kohler and Langer, 2021]) if it takes the form as

$$g = f_q \circ f_{q-1} \circ \cdots \circ f_1 \tag{5}$$

where $f_j : (a_j, b_j)^{d_j} \to (a_{j+1}, b_{j+1})^{d_{j+1}}$, $d_0 = \mathfrak{p} + \mathfrak{d}$ and $d_{q+1} = D$. Denote $f_j = (f_j^{(1)}, \ldots, f_j^{(d_{j+1})})$ as the components of $f_j$, let $t_j$ be the maximal number of variables on which each of the $f_j^{(i)}$ depends and let $f_j^{(i)} \in \mathcal{H}^{\beta_j}((a_j, b_j)^{t_j}, K)$ (see Section 2.4.1 for the definition of the Hölder class $\mathcal{H}^\beta$). A composite structure is very general which includes smooth functions and additive structure as special cases. In addition, in the next section, we show the class of conditional distributions $\{Q_{G_*}(\cdot|X = x) : x \in \mathbb{R}^{\mathfrak{p}}, G_* \in \mathcal{G}\}$ induced by the composite structure is broad.

**Assumption 2** (composite structure ). *Denote $\mathcal{G} = \mathcal{G}(q, \boldsymbol{d}, \boldsymbol{t}, \boldsymbol{\beta}, K)$ as a collection of functions of form (5), where $\boldsymbol{d} = (d_0, \ldots, d_{q+1})$, $\boldsymbol{t} = (t_0, \ldots, t_{q+1})$, and $\boldsymbol{\beta} = (\beta_0, \ldots, \beta_{q+1})$. We regard $(q, \boldsymbol{d}, \boldsymbol{t}, \boldsymbol{\beta}, K)$ as constants in our setup, and assume that the true generator $G_*(\cdot, x)$ as in (2) belongs to $\mathcal{G}$, for all $x \in \mathcal{X}$. Additionally, we assume $\||G_*|_\infty\|_\infty \leq K$.*

$$\widetilde{\beta}_j = \beta_j \prod_{l=j+1}^q (\beta_l \wedge 1), \qquad j_* = \operatorname*{argmax}_{j \in \{0, \ldots, q\}} \frac{t_j}{\widetilde{\beta}_j}, \qquad \beta_* = \widetilde{\beta}_{j_*}, \qquad t_* = t_{j_*}.$$

*The quantities $t_*$ and $\beta_*$ are called intrinsic dimension and smoothness of $G_*$ (or of $\mathcal{G}$).*

**Remark 1** (Strength of the Composite Structure). *The expression $(a_j, b_j) \subset [-K, K]$ can be intuitively visualized by setting $a_j = -K$ and $b_j = K$. To illustrate the impact of intrinsic dimensionality and smoothness, consider a function $f : \mathbb{R}^d \to \mathbb{R}$ defined as $f(x) = f_1(x_1) + \ldots + f_d(x_d)$, where $x = (x_1, \ldots, x_d)$ and $f_j \in \mathcal{H}^\beta((-K, K), K)$ for $j = 1, \ldots, d$. While $f \in \mathcal{H}^\beta((-K, K)^d, K)$, its intrinsic dimension is $t_* = 1$ with intrinsic smoothness $\beta$. This mitigates the curse of dimensionality.*

**Assumption 3.** *Let $\mathcal{M}_*$ be the closure of $G_*(\mathcal{Z}, \mathcal{X})$. We assume that $\mathcal{M}_*$ does not have an interior point, and $\operatorname{reach}(\mathcal{M}_*) = \mathsf{r}_*$ with $\mathsf{r}_* > 0$.*

Assumption 2 permits low intrinsic dimensionality within the learnable function class. Assumption 3 imposes the strong identifiability condition necessary for efficient estimation, as seen in manifold literature ([Aamari and Levrard, 2019]; [Tang and Yang, 2023]).

Given two conditional densities $p_1(\cdot|x), p_2(\cdot|x)$ and $\mu_X^*$ denoting the density of $X$, we use integrated distances for a measure of evaluation. With a slight abuse of notation, we denote $d_1(p_1, p_2) = \mathbb{E}_X[d_1(p_1(\cdot|x), p_2(\cdot|x))]$ and $d_H(p_1, p_2) = \mathbb{E}_X[d_H(p_1(\cdot|x), p_2(\cdot|x))]$, where $d_1$ and

$d_H$ represent the $L_1$ and the Hellinger distance as $d_1(p_1(\cdot|x), p_2(\cdot|x)) = \int |p_1(y|x) - p_2(y|x)| \, dy$ and $d_H(p_1, p_2) = (\int \int [\sqrt{p_1(y|x)} - \sqrt{p_2(y|x)}]^2 \, dy)^{1/2}$ respectively. Denote $\mathcal{N}(\delta, \mathcal{F}, d)$ and $\mathcal{N}_{[]}(\delta, \mathcal{F}, d)$ as covering and bracketing numbers of the function class $\mathcal{F}$ with respect to the (pseudo)-metric $d$.

We first present Lemma 1, which establishes the bracketing entropy of the functional class $\mathcal{P}$ with respect to Hellinger distance in terms of the covering entropy of the search class $\mathcal{F}$. This enables us to transfer the entropy control of the individual components $\mathcal{F}$ and $\sigma$ to the entire $\mathcal{P}$.

**Lemma 1.** *Let $\mathcal{F}$ be class of functions from $\mathcal{Z} \times \mathcal{X}$ to $\mathbb{R}^D$ such that $\||g|_\infty\|_\infty \leq K$ for every $g \in \mathcal{F}$. Let $\mathcal{P} = \{P_{g,\sigma} : g \in \mathcal{F}, \sigma \in [\sigma_{\min}, \sigma_{\max}]\}$ with $\sigma_{\min} \leq 1$. Then, there exist constants $c = c(\sigma_{\max}, K, D)$ and $C = C(\sigma_{\max}, K, D)$ and $\delta_* = \delta_*(D)$ such that for every $\delta \in (0, \delta_*]$,*

$$\log \mathcal{N}_{[]}(\delta, \mathcal{P}, d_H) \leq \log \mathcal{N}(c\sigma_{\min}^{D+3}\delta^4, \mathcal{F}, \||\cdot|_\infty\|_\infty) + \log\left(\frac{C}{\sigma_{\min}^{D+2}\delta^4}\right), \tag{6}$$

The proof of Lemma 1 is provided in the Appendix E. Theorem 1 presents the convergence rate of the sieve-MLE to the true distribution (see Appendix F for the proof).

**Theorem 1.** *Let $\mathcal{F}, \mathcal{P}, \sigma_{\min}$ and $\delta_* = \delta_*(D)$ be given as in Lemma 1, and $n \geq 1$. Suppose that $\log \mathcal{N}(\delta, \mathcal{F}, \||\cdot|_\infty\|_\infty) \leq \xi\{A + 1 \vee \log \delta^{-1}\}$ for every $\delta \in (0, \delta_*]$ and some $A, \xi > 0$. Suppose that there exists a $G \in \mathcal{F}$ and some $\delta_{\text{approx}} \in (0, \delta_*]$ such that $\||G - G_*|_\infty\|_\infty \leq \delta_{\text{approx}}$. Furthermore, suppose that $s \geq 1$, $A \geq 1$, $\sigma_{min} \leq 1$, $\delta_{\text{approx}} \leq 1$ and $\sigma_* \in [\sigma_{\min}, \sigma_{\max}]$. Then*

$$P_*\left(d_H(\widehat{p}, p_*) > \varepsilon_n^*\right) \leq 5e^{-C_1 n \varepsilon_n^{*2}} + C_2 n^{-1} \tag{7}$$

*provided that $\eta_n \leq n\varepsilon_n^{*2}/6$ and $\varepsilon_n^* \leq \sqrt{2}\delta_*$, where*

$$\varepsilon_n^* = C_3\left(\sqrt{\frac{\xi\{A + \log(n/\sigma_{\min})\}}{n}} \vee \frac{\delta_{\text{approx}}}{\sigma_*}\right), \tag{8}$$

*$C_1$ is an absolute constant, $C_2 = C_2(D)$ and $C_3 = C_3(D, K, \sigma_{\max})$.*

The outlined rate has two components: the statistical component, expressed as an upper bound to the metric entropy of $\mathcal{F}$, and the approximation component, denoted as $\delta_{\text{approx}}$. The statistical error is quantified by measuring the complexity of the class $\mathcal{P}$, as formulated in Lemma 1. The approximation error is assessed through the ability of the provided function class to approximate the true distribution.

## 2.2 NEURAL NETWORK CLASS

We model $G_*(\cdot, \cdot)$ using a deep neural network. More specifically, we parameterize the true generator $G_*$ with a deep neural neural architecture $(L, \mathbf{r})$ of the form

$$f : \mathbb{R}^{r_0} \to \mathbb{R}^{r_{L+1}}, \qquad z \mapsto f(z) = W_L \rho_{v_L} W_{L-1} \rho_{v_{L-}} \dots W_1 \rho_{v_1} W_0 z, \tag{9}$$

where $W_j \in \mathbb{R}^{r_{j+1} \times r_j}$, $v_j \in \mathbb{R}^{r_j}$, $\rho_{v_j}(\cdot) = \text{ReLU}(\cdot - v_j)$ and $\mathbf{r} = (r_0, \dots, r_{L+1}) \in \mathbb{N}^{L+2}$. The constant $L$ is the number of hidden layers and $r = (r_0, \dots, r_{L+1})$ represents the number of nodes in each layer.

We define the **sparse** neural architecture class $\mathcal{F}_s(L, \mathbf{r}, s, B, K)$ as set of functions of form (9) satisfying

$$\max_{0 \leq j \leq L} |W_j|_\infty \vee |v_j|_\infty \leq B, \qquad \sum_{j=1}^{L} |W_j|_0 + |v_j|_0 \leq s, \qquad \||f|_\infty\|_\infty \leq K,$$

with $r_0 = \mathfrak{d} + \mathfrak{p}$ and $r_{L+1} = D$, where $|\cdot|_0$ and $|\cdot|_\infty$ stand for the $L^0$ and $L^\infty$ vector norms, and $\||f|_\infty\|_\infty = \sup_{x \in \mathbb{R}^{r_0}} \max_{i=1,\dots,D} |f_i(x)|$, $s$ is sparsity parameter and $K$ is functional bound.

The **fully connected** neural architecture class $\mathcal{F}_c = \mathcal{F}_c(L, \mathbf{r}, B, K)$ is set of functions of form (9) satisfying

$$\max_{0 \leq j \leq L} |W_j|_\infty \vee |v_j|_\infty \leq B, \qquad \||f|_\infty\|_\infty \leq K.$$

Both classes $\mathcal{F}_s$ and $\mathcal{F}_c$ for the deep generator will be considered in our analysis of the resulting sieve maximum likelihood estimator. We denote the corresponding sieve-MLE as $\widehat{p}_s$ and $\widehat{p}_c$, respectively. When we use $r$ instead of $\mathbf{r}$, it refers to $r_1 = \ldots = r_L = r$ along with $r_0 = \mathfrak{d} + \mathfrak{p}$ and $r_{L+1} = D$.

We can simplify and visualize the result stated in Theorem 1 in both cases: when the sieve-MLE is obtained with optimization performed over the class $\mathcal{F}_s$ and $\mathcal{F}_c$. To fulfill the conditions stated in the Theorem 1, we need to establish entropy bounds for these function classes, $\mathcal{F}_s$ and $\mathcal{F}_c$, and gain insight into their approximation capabilities for the composite structure class described in Assumption 2.

For the sparse neural architecture class $\mathcal{F}_s(L, r, s, K)$, the entropy, formally stated as Proposition 1 in Ohn and Kim (2019), is bounded as follows.

$$\log \mathcal{N}(\delta, \mathcal{F}_s, \|| \cdot |_\infty\|_\infty) \lesssim sL \{\log(BLr) + \log \delta^{-1}\}. \tag{10}$$

From an entropy perspective, the fully connected neural architecture class $\mathcal{F}_c(L, r, B, K)$ can be viewed as $\mathcal{F}_s$ without any sparsity constraint, meaning $s \asymp r^2 L$. Therefore, we have

$$\log \mathcal{N}(\delta, \mathcal{F}_c, \|| \cdot |_\infty\|_\infty) \lesssim L^2 r^2 \{\log(BLr) + \log \delta^{-1}\}. \tag{11}$$

The approximation properties of the sparse and fully connected network are provided in Lemma 4.1 and Lemma 4.2 of the Appendix K, respectively.

Having established the essential components for $\mathcal{F}_c$ in (11) and Lemma 4.2, and for $\mathcal{F}_s$ in (10) and Lemma 4.1, respectively, we can simplify Theorem 1 and state Corollary 1.

**Corollary 1.** *Suppose that Assumptions 1 and 2 hold, and $\sigma_* \in [\sigma_{\min}, \sigma_{\max}]$ with $\sigma_{\min} \leq 1$ and $\sigma_{\max} < \infty$. Moreover, assume that the noise $\sigma_*$ decays at rate $\alpha$, i.e., $\sigma_* \asymp n^{-\alpha}$, and $\sigma_{\min} = n^{-\gamma}$ for some $\gamma \geq \alpha \geq 0$. Then, for every $\delta_{\text{approx}} \in [0, 1]$, the following holds:*

1. *Let $\mathcal{F}_s = \mathcal{F}_s(L, r, s, B, K)$ with $\delta_* = \delta_*(D)$ be as given in Lemma 1, and $L \asymp \log \delta_{\text{approx}}^{-1}$, $r \asymp \delta_{\text{approx}}^{-t_*/\beta_*}$, $s \asymp \delta_{\text{approx}}^{-t_*/\beta_*} \log \delta_{\text{approx}}^{-1}$, $B \asymp \delta_{\text{approx}}^{-1}$. Then the sieve MLE $\widehat{p}_s$ satisfies (7) with $\varepsilon_n^*$ as in (8) with $\xi = \delta_{\text{approx}}^{-t_*/\beta_*} \log^2(\delta_{\text{approx}}^{-1})$ and $A = \log^2(\delta_{\text{approx}}^{-1})$ provided that $\eta_n \leq n \varepsilon_n^{*2}/6$ and $\varepsilon_n^* \leq \sqrt{2}\delta_*$.*

2. *Let $\mathcal{F}_c = \mathcal{F}_c(L, r, B, K)$ with $\delta_* = \delta_*(D)$ be as given in Lemma 1, and $L \asymp \log \delta_{\text{approx}}^{-1}$, $r \asymp \delta_{\text{approx}}^{-t_*/2\beta_*}$, $B \asymp \delta_{\text{approx}}^{-1}$. Then the sieve MLE $\widehat{p}_c$ satisfies (7) with $\varepsilon_n^*$ as in (8) with $\xi = \delta_{\text{approx}}^{-t_*/\beta_*} \log^2(\delta_{\text{approx}}^{-1})$ and $A = \log^2(\delta_{\text{approx}}^{-1})$ provided that $\eta_n \leq n \varepsilon_n^{*2}/6$ and $\varepsilon_n^* \leq \sqrt{2}\delta_*$.*

*In particular, choosing $\delta_{\text{approx}} := (\sigma_*^2/n)^{\beta_*/(2\beta_* + t_*)}$ minimizes $\varepsilon_n^* \asymp \sqrt{\xi \{A + \log(n/\sigma_{\min})\}/n} \vee \delta_{\text{approx}}/\sigma_*$, and gives*

$$\varepsilon_n^* \asymp n^{-\frac{\beta_* - t_*\alpha}{2\beta_* + t_*}} \log^2(n). \tag{12}$$

**Remark 2.** *The convergence rate in (12) illustrates the influence of intrinsic dimensionality, smoothness, and noise level on the estimation process. Note that $\alpha$ is upper bounded as $\varepsilon_n^* \leq \sqrt{2}\delta_*(D)$. For large values of $\alpha$, estimation of $G_*$ is inherent difficult as the data is very close on the singular support. To address this, a small noise injection, as described in Corollary 2, can smooth the estimation and ensure consistency.*

The proof of Corollary 1 is provided in Appendix G. For the composite structural class $\mathcal{G}$, the effective smoothness is denoted by $\beta_*$, and the dimension is $t_*$. This effectively mitigates the curse of dimensionality. The convergence rate at (12) also recovers the optimal rate when $q = 1$ and $\alpha = 0$, and there is a small lag of polynomial factor $t_*\alpha/(2\beta_* + t_*)$ when $\alpha > 0$ (Norets and Pati, 2017). This lag arises due to the presence of full-dimensional noise in the response observation $Y$. Note that when the noise is small, that is $\alpha$ is large, achieving a sharp estimation of $p_*$ requires an equally accurate estimate of $G_*$. This can be quite challenging. Our practically tractable approach attempts to address this without initially estimating the singular support.

## 2.3 WASSERSTEIN CONVERGENCE OF THE INTRINSIC (CONDITIONAL) DISTRIBUTIONS

Using Wasserstein distance as a metric for distributions $Q_g$ is meaningful due to their singularity in ambient space: when $\mathfrak{d} < D$, the conditional distribution is singular with respect to the Lebesgue measure on $\mathbb{R}^D$.

The integrated Wasserstein distance, for $r \geq 1$, between $P_1(\cdot|X)$ and $P_2(\cdot|X)$ is defined as

$$W_r\left(P_1, P_2\right) = \mathbb{E}_X\left[\inf_{\beta \in \Gamma(P_1, P_2)} \left(\mathbb{E}_{(U_1, U_2) \sim \beta}\left[|U_1 - U_2|_r^r\right]\right)^{1/r}\right],$$

where $\Gamma(P_1, P_2)$ is the set of all couplings between $P_1$ and $P_2$ that preserves the two marginals. The (dual) representation of this norm, $W_r(P_1, P_2) = \mathbb{E}_X\left[\sup_{\|f\|_{\mathrm{Lip}_r} \leq 1}\left\{\mathbb{E}_{P_1}[f] - \mathbb{E}_{P_2}[f]\right\}\right]$ (Villani et al., 2009) with $\|\cdot\|_{\mathrm{Lip}_r}$ denoting the $r$-Lipschitz norm, is particularly useful in our proofs.

**Theorem 2.** *Suppose that Assumption 3 holds. If $d_H(p_{g,\sigma}, p_*) \leq \varepsilon$ holds for some $\varepsilon \in [0, 1]$ and some $p_{g,\sigma} \in \mathcal{P}$, then we have*

$$W_1(Q_g, Q_*) \leq C\left(\varepsilon + \sigma_* \sqrt{\log \varepsilon^{-1}}\right),$$

*where $C = C(D, K, \mathsf{r}_*)$ depends only on $(D, K, \mathsf{r}_*)$.*

The proof of Theorem 2 is provided in Appendix H. Theorem 2 guarantees that $W_1\left(\widehat{Q}_{\widehat{g}}, Q_*\right) \lesssim_{\log} d_H(\widehat{p}, p_*) + \sigma_*$, where $\lesssim_{\log}$ represents less than or equal up to a logarithmic factor of $n$. Following from Corollary 1, the Wasserstein convergence rate, $n^{-(\beta_* - t_* \alpha)/(2\beta_* + t_*)} \log^2(n) \vee \sigma_* \log^{1/2}(n)$, comprises two components: the convergence rate in the Hellinger distance and the standard deviation of the true noise sequence. It is noteworthy that the first expression is influenced by the variance of noise by the factor $\alpha$. When $\alpha$ is very small, indicating that the data $Y_j$ lies very close to the manifold, the second expression $n^{-\alpha}$ in the overall rate dominates. Intuitively, this phenomenon arises from the underlying structural challenges in related manifold estimation problems with noisy data, as discussed by Genovese et al. (2012). To address this issue, we propose a data perturbation strategy by transforming the data $\{(Y_j, X_j)\}_{j=1}^n$ into $\{(\widetilde{Y}_j, X_j)\}_{j=1}^n$, where $\widetilde{Y}_j = Y_j + \epsilon_j$ and $\epsilon_j \sim \mathsf{N}\left(0_D, n^{-\beta_*/(\beta_* + t_*)} I_D\right)$. The resulting estimation error bound is summarized below, whose proof is provided in Appendix I.

**Corollary 2.** *Suppose that Assumption 1, 2, and 3 hold, and $\sigma_* \in [\sigma_{\min}, \sigma_{\max}]$ with $\sigma_* = n^{-\alpha}$ and $\sigma_{\min} = n^{-\gamma}$ for some $0 \leq \alpha \leq \gamma$. Then for each of the network architecture classes (sparse and fully connected) with the network parameters specified in Corollary 1, the sieve MLE $\widehat{p}_{per}$ and $\widehat{Q}_{per}$ based on the perturbed data $\{(\widetilde{Y}_j, X_j)\}_{j=1}^n$ satisfies*

$$P_*\left[W_1\left(\widehat{Q}_{per}, Q_*\right) \geq \left(\varepsilon_n^* + \sigma_* \sqrt{\log((\varepsilon_n^*)^{-1})}\right)\right] \lesssim 5e^{-C_1 n \varepsilon_n^{*2}} + \frac{C_2}{n}$$

*where $\varepsilon_n^*$ can be chosen such that*

$$\varepsilon_n^* + \sigma_* \sqrt{\log((\varepsilon_n^*)^{-1})} \asymp \begin{cases} n^{-\frac{\beta_* - t_* \alpha}{2\beta_* + t_*}} \log^2(n), & \text{if } \alpha < \beta_*/\{2(\beta_* + t_*)\}, \\ n^{-\frac{\beta_*}{2(\beta_* + t_*)}} \log^2(n), & \text{otherwise.} \end{cases} \quad (13)$$

## 2.4 CHARACTERIZATION OF THE LEARNABLE DISTRIBUTION CLASS

Section 2.2 focuses on the true generator $G_*$ within the class of functions with composite structures. In this subsection, we show that such a conditional distribution class achieved by the push-forward map $G_*$ is broad and includes many existing distribution classes for $Q_*$ as special cases.

### 2.4.1 SMOOTH CONDITIONAL DENSITY

For $\beta > 0$, let $\mathcal{H}^\beta(D, M)$ be the class of all $\beta$-Hölder functions $f : D \subset \mathbb{R}^\mathfrak{d} \to \mathbb{R}$ with $\beta$-Hölder norm bounded by $M > 0$. Let $\mathcal{H}^\beta(D) = \cup_{M>0} \mathcal{H}^\beta(D, M)$. See Appendix B for their formal definitions.

**Lemma 2.** *Suppose that (i) $\mathcal{Z} \times \mathcal{X}$ and $\mathcal{Y}$ are uniformly convex and (ii) $p_Z \in \mathcal{H}^{\beta_Z}(\mathcal{Z})$, $\mu_X^* \in \mathcal{H}^{\beta_X}(\mathcal{X})$ and $q_* \in \mathcal{H}^{\beta_Q}(\mathcal{Y})$ for some $\beta_Z, \beta_X, \beta_Q > 0$ and are bounded above and below. Then, there exists a map $g(\cdot, \cdot) : \mathcal{Z} \times \mathcal{X} \to \mathcal{Y}$ such that $Q_*(\cdot|\cdot) = Q_g$ and $g \in \mathcal{H}^{\beta_{\min} + 1}(\mathcal{Z} \times \mathcal{X})$, where $\beta_{\min} = \min\{\beta_Z, \beta_X, \beta_Q\}$.*

Lemma 2 establishes that the learnable distribution class includes Hölder-smooth functions with smoothness parameter $\beta_{\min}$ and intrinsic dimension $\mathfrak{d}$. As a result, following Corollary 1, the convergence rate for density estimation is given by $\varepsilon_n^* \asymp n^{-(\beta_{\min}+1-\mathfrak{d}\alpha)/(2\beta_{\min}+2+\mathfrak{d})}$. A push-forward map is a transport map between two distributions. The well-established regularity theory of transport map in optimal transport is directly applicable here [see Villani et al. (2009) and Villani (2021)]. The proof of Lemma 2 is based on Theorem 12.50 of Villani et al. (2009) and Caffarelli (1996), which establishes the regularity of this transport map and its existence follows from Brenier (1991). When $p_Z$ is selected as a well-behaved parametric distribution, the regularity of the transport map is determined by the smoothness of both $\mu_X^*$ and $Q_*$. For a more detailed discussion on this, please refer to Appendix C.

### 2.4.2 A BROADER CONDITIONAL DISTRIBUTION CLASS WITH SMOOTHNESS DISPARITY

In Appendix L, we present a novel approximation result for the function class exhibiting *smoothness disparity* in Theorem 5. This new result facilitates the study of theoretical properties of estimators when the generator $G_* \in \mathcal{H}_{\mathfrak{d},\mathfrak{p}}^{\beta_Z,\beta_X}(\mathcal{Z},\mathcal{X},K)$. Note that such a function class defined in (16) in Appendix L is much broader compared to the smoothness class in Section 2.4.1 as $Z$ and $X$ do not have to be jointly smooth and it allows for smoothness disparity among them. The subsequent Theorem 3 combines our approximation result with (11) and enables us to specialize Theorem 1 to this class (see Appendix J for the proof).

**Theorem 3.** *Let $G_* \in \mathcal{H}_{\mathfrak{d},\mathfrak{p}}^{\beta_Z,\beta_X}(\mathcal{Z},\mathcal{X},K)$. Suppose that Assumption 1 holds and $\sigma_* \in [\sigma_{\min},\sigma_{\max}]$ with $\sigma_{\min} \leq 1$ and $\sigma_{\max} < \infty$. Moreover, we assume $\sigma_* \asymp n^{-\alpha}$, and $\sigma_{\min} = n^{-\gamma}$ for some $0 \leq \alpha \leq \gamma \leq (\beta_Z^{-1}\mathfrak{d} + \beta_X^{-1}\mathfrak{p})^{-1}$. Then, for every $\delta_{\mathrm{approx}} \in [0,1]$, we have: Let $\mathcal{F}_s = \mathcal{F}_s(L,r,s,1,K)$ with $L \asymp \log \delta_{\mathrm{approx}}^{-1}$, $r \asymp \delta_{\mathrm{approx}}^{-(\beta_Z^{-1}\mathfrak{d}+\beta_X^{-1}\mathfrak{p})}$, $s \asymp \delta_{\mathrm{approx}}^{-(\beta_Z^{-1}\mathfrak{d}+\beta_X^{-1}\mathfrak{p})} \log \delta_{\mathrm{approx}}^{-1}$. Then the sieve MLE $\widehat{p}_s$ satisfies (7) with the rate outlined in (8) with $\xi = \delta_{\mathrm{approx}}^{-(\beta_Z^{-1}\mathfrak{d}+\beta_X^{-1}\mathfrak{p})} \log^2 \delta_{\mathrm{approx}}^{-1}$ and $A = \log^2 \delta_{\mathrm{approx}}^{-1}$, provided that $\eta_n \leq n\varepsilon_n^{*2}/6$. In particular, choosing $\delta_{\mathrm{approx}} := \left(\sigma_*^2/n\right)^{1/\left(2+\beta_Z^{-1}\mathfrak{d}+\beta_X^{-1}\mathfrak{p}\right)} \leq 1$ minimizes $\varepsilon_n^* \asymp \sqrt{\xi\left\{A + \log\left(n/\sigma_{\min}\right)\right\}/n} \vee \delta_{\mathrm{approx}}/\sigma_*$, and gives*

$$\varepsilon_n^* \asymp n^{-\frac{1-\alpha(\beta_Z^{-1}\mathfrak{d}+\beta_X^{-1}\mathfrak{p})}{2+\beta_Z^{-1}\mathfrak{d}+\beta_X^{-1}\mathfrak{p}}} \log^2(n). \tag{14}$$

The proof of Theorem 3 is provided in Appendix J. In the special case when $\alpha = 0$ and $\mathfrak{d} = D$, our convergence rate in (14) recovers the minimax optimal rate for conditional density estimation based on kernel smoothing, as established in Li et al. (2022).

### 2.4.3 CONDITIONAL DISTRIBUTION ON MANIFOLDS

In this part, we extend Lemma 2 and provide the existence of the generator when the conditional distribution is supported on a compact manifold with dimension $\mathsf{d}_* \leq D$. Due to space constraints, we provide only a sketched proof here; the detailed proof can be found in Appendix D. Specifically, we first present arguments for the existence of the generator when $\mathcal{Y}$ is covered by a single chart. We then extend this to the multiple chart case using the technique of partition of unity.

In the simpler case when there exists a single $(\mathcal{Y},\varphi)$ covering $\mathcal{Y}$, where $\varphi : \mathcal{B}_1(0_{\mathsf{d}_*}) \to \mathcal{Y}$ is a homeomorphism, we assume $\varphi \in \mathcal{H}^{\beta_{\min}+1}$. In this case, we use the change of variable formula to transfer the measure on $\mathcal{B}_1(0_{\mathsf{d}_*})$ (unit ball in $\mathbb{R}^{\mathsf{d}_*}$) from $\mathcal{Y}$. Following Lemma 2, we can find a transport map $g \in \mathcal{H}^{\beta_{\min}}$ mapping from $\mathcal{Z} \times \mathcal{X}$ to $\mathcal{B}_1(0_{\mathsf{d}_*})$. The map $g \circ \varphi$ then serves as our generator.

In the general case where the compact manifold $\mathcal{Y}$ needs to be covered by multiple charts, demonstrating the existence of a transport or push-forward map is challenging because $\mathcal{Y}$ is not uniformly convex. Suppose that $\{(U_k,\varphi_k)\}_{k=1}^{K}$ forms a cover of $\mathcal{Y}$. Due to the compactness of $\mathcal{Y}$, the number of charts $K$ is finite. Analogous to the single chart scenario, we first construct $g_k \circ \varphi_k$ to transport the measure on each chart. We then patch these local transport maps together to construct a global transport map; see Appendix D for full details. As a result, following Corollary 1, the convergence rate for density estimation shall be given by $\varepsilon_n^* \asymp n^{-(\beta_{\min}-\mathfrak{d}\alpha)/(2\beta_{\min}+\mathfrak{d})}$.

## 3  NUMERICAL RESULTS

In this section, we present numerical experiments to validate and complement our theoretical findings using two synthetic dataset examples. These experiments cover a range of scenarios, including full-dimensional cases as well as benchmark examples involving manifold-based data. Additionally, we provide a real data example to further enrich our experimentation and validation process. It is worth noting that, although not significant, the computational cost of fitting a conditional generative model is higher compared to fitting an unconditional one, as the input dimension of the deep neural network (DNN) is $\mathfrak{p} + \mathfrak{d}$ rather than just $\mathfrak{d}$.

**Learning algorithm to compute sieve MLE.**  For the computational algorithm, we adopt a common conditional variational auto-encoder (VAE) architecture to maximize the following log-likelihood term:$\sum_{j=1}^{n} \mathcal{L}_{\mathrm{VAE}}(g, \sigma, \phi; Y_j, X_j)$, where

$$\mathcal{L}_{\mathrm{VAE}}(g, \sigma, \phi; y, x) = \log\left(\frac{p_{g,\sigma}(y, x, z)}{q_\phi(Z|y, x)}\right).$$

The variational distribution $q_\phi(Z|y, x)$ is chosen as the standard normal family $\mathsf{N}(\mu_\phi(y, x), \Sigma_\phi(y, x))$.

We examine two classes of datasets: (i) full-dimensional response and (ii) response residing on a low-dimensional manifold. The first highlights the generality of our proposed approach, while the second underscores its efficiency in terms of the Wasserstein metric and validates the small noise perturbation strategy outlined in Corollary 2.

**Simulation from full dimension distribution**. We use the following models for data generation.

- **FD1** : $Y = \mathbb{I}_{\{U < 0.5\}}\, \mathsf{N}\left(-X, 0.25^2\right) + \mathbb{I}_{\{U > 0.5\}}\, \mathsf{N}\left(X, 0.25^2\right)$; $U \sim \mathrm{Unif}(0, 1)$, $X \sim \mathsf{N}(3, 1)$.

- **FD2** : $Y = X_1^2 + e^{(X_2 + X_3/3)} + \sin(X_4 + X_5) + \varepsilon$; $\{X_j\}_{j=1}^5 \overset{i.i.d}{\sim} \mathsf{N}(0, 1)$, $\varepsilon \sim \mathsf{N}(0, 1)$.

- **FD3** : $Y = X_1^2 + e^{(X_2 + X_3/3)} + X_4 - X_5 + 0.5\left(1 + X_2^2 + X_5^2\right) \times \varepsilon$; $\{X_j\}_{j=1}^5 \overset{i.i.d}{\sim} \mathsf{N}(0, 1)$, $\varepsilon \sim \mathsf{N}(0, 1)$.

These are examples of a mixture model, an additive noise model, and a multiplicative noise model, respectively. The neural architecture for both the encoder and decoder consists of two deep layers, i.e., $L = 2$. The hyperparameters are as follows: $r_{\mathrm{enc}} = (\mathfrak{p} + 1, 10, 10)$ for $\mu_\phi$ and $\Sigma_\phi$, and $r_{\mathrm{dec}} = (10 + \mathfrak{p}, 10, 1)$ for $g$. The sample size used for simulation is 5000, with a training-to-testing ratio of $4 : 1$. We employ a batch size of 64 with a learning rate of $10^{-3}$.

We compare the sieve MLE with CKDE (Hall et al., 2004) and FlexCode proposed by Izbicki and Lee (2017). To evaluate their performance, we compute the mean squared error (MSE) for both the mean and the standard deviation. We use Monte Carlo approximation to compute the mean and standard deviation for the sieve MLE, and numerical integration for CKDE and Flexcode. This evaluation strategy resembles that implemented by Zhou et al. (2022). Table 1 summarizes the findings.

Table 1: MSE for the estimated conditional mean and the standard deviation.

|  |  | Sieve MLE | CKDE | FlexCode |
|---|---|---|---|---|
| FD1 | MEAN | **0.0379** $\pm$ 0.0170 | 1.0053 $\pm$ 0.1004 | 1.1660 $\pm$ 0.1076 |
|  | SD | **0.0280** $\pm$ 0.0045 | 0.9887 $\pm$ 0.0347 | 1.2000 $\pm$ 0.0126 |
| FD2 | MEAN | **0.1943** $\pm$ 0.0427 | 0.2640 $\pm$ 0.0515 | 0.3954 $\pm$ 0.0571 |
|  | SD | **0.2843** $\pm$ 0.0093 | 0.2853 $\pm$ 0.0213 | 5.8278 $\pm$ 0.1607 |
| FD3 | MEAN | **0.2337** $\pm$ 0.0453 | 0.2967 $\pm$ 0.0537 | 1.3419 $\pm$ 0.1087 |
|  | SD | 1.6394 $\pm$ 0.0861 | **0.6334** $\pm$ 0.0460 | 11.4898 $\pm$ 0.1559 |

Note that the sieve MLE outperforms all other methods in all scenarios except for the MSE(SD) for the FD3 dataset. However, for the FD3 dataset, we found that as the training sample size increases further, the MSE(SD) of the sieve MLE achieves performance increasingly comparable to CKDE.

**Simulation from distributions on manifolds.** We consider two examples of manifolds with an intrinsic dimension $\mathfrak{d} = 1$, while the ambient dimension is $D = 2$.

- **M1** : $Y = G_*(Z, U) + \varepsilon$, $G_* = (G_*^{(1)}, G_*^{(2)})$, $G_*^{(1)} = \mathbb{I}_{\{U < 0.5\}}(1 - \cos(Z)) + \mathbb{I}_{\{U > 0.5\}}\cos(Z)$,
  $G_*^{(2)} = \mathbb{I}_{\{U < 0.5\}}(0.5 - \sin(Z)) + \mathbb{I}_{\{U > 0.5\}}\sin(Z)$; $Z \sim \text{Unif}(0, \pi)$, $U \sim \text{Unif}(0, 1)$.

- **M2** : $Y = G_*(Z, U) + \varepsilon$, $G_* = \left(G_*^{(1)}, G_*^{(2)}\right)$, $G_*^{(1)} = \mathbb{I}_{\{U < 0.5\}}\cos(Z) + \mathbb{I}_{\{U > 0.5\}}2\cos(Z)$,
  $G_*^{(2)} = \mathbb{I}_{\{U < 0.5\}}0.5\sin(Z) + \mathbb{I}_{\{U > 0.5\}}\sin(Z)$; $Z \sim \text{Unif}(0, 2\pi)$, $U \sim \text{Unif}(0, 1)$.

The manifold $M_1$ consists of two moons. The manifold $M_2$ comprises ellipses, with conditions distinguishing the inner and outer confocal ellipses. The noise sequence follows a two-dimensional centered Gaussian distribution, $\varepsilon \sim \text{N}(0_2, \sigma_*^2 I_2)$. We investigated this setup across various noise variances $\sigma_*^2$. Our neural architecture employed $r_{\text{enc}} = (\mathfrak{p} + 2, 100, 100, 2)$ for $\mu_\phi$ and $\Sigma_\phi$, and $r_{\text{dec}} = (2 + \mathfrak{p}, 100, 100, 2)$ for $g$. We utilized a sample size of 5000 for simulation, with a training-to-testing ratio of $4 : 1$. A batch size of 100 was employed, with a learning rate of $10^{-3}$. We

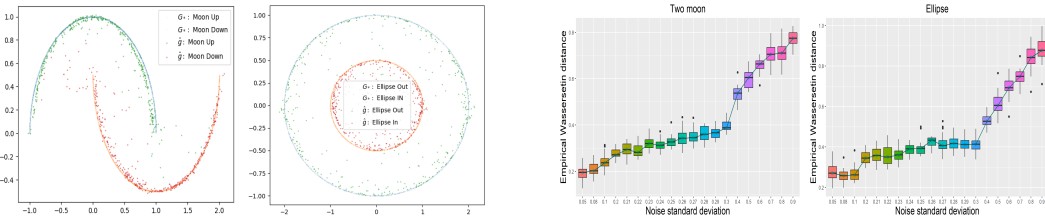

Figure 1: Generated samples from manifold $M_1$ and $M_2$ are displayed in the left panel. The right panel shows box plots for the empirical Wasserstein distance at different noise levels $\sigma_*$.

computed the empirical $W_1$ distance using the algorithm proposed by Cuturi (2013) to evaluate the performance. The right panel of Figure 1 presents the boxplots of $W_1$ between the true and learned distribution for $M_1$ and $M_2$ across 20 repetitions. The left panel highlights the following general behaviors:

- When $\alpha$ is small and close to zero, the noise variance is large, making estimation challenging due to the singularity of the true data distribution.
- When $\alpha$ is large, the noise variance is small, and the perturbed data facilitates efficient estimation.

This observed pattern, as emphasized in Corollary 2, closely aligns with the results achieved in (13). An additional numerical experiment on real data has been performed and can be found in Appendix A.1.

## 4 DISCUSSION

We investigated statistical properties of a likelihood-based conditional deep generative model for distribution regression in a scenario where the response variable is situated in a high-dimensional ambient space but is centered around a potentially lower-dimensional intrinsic structure. Our analysis established favorable rates in both the Hellinger and Wasserstein metrics which are dependent on only the intrinsic dimension of the data. Our theoretical findings show that the conditional deep generative models can circumvent the curse of dimensionality for high-dimensional distribution regression. To the best of our knowledge, our work is the first of its kind.

Given the novelty of emerging statistical methodologies with intricate structural considerations in the study of deep generative models, there exist numerous paths for future exploration. Among these potential directions, we are particularly interested in investigating controllable generation via penalized optimization methods, studying statistical properties of deep generative models trained via matching flows, as well as delving into the hypothesis testing problem within the framework of deep generative models, among others. Another interesting direction is to explore residual neural network structure for modeling time series of distributions with interesting temporal dependence structures.

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
