# OpenReview forum: "A Likelihood Based Approach to Distribution Regression Using Conditional Deep Generative Models"
_ICLR.cc/2025/Conference — Submitted to ICLR 2025_

### Official Review · Reviewer_VL1U · 2024-10-17

**Soundness:** 3
**Presentation:** 2
**Contribution:** 3
**Rating:** 8
**Confidence:** 3

**Summary:**

This paper aims at providing statistical bounds for conditional generative models under various smoothness assumptions. The overall goal is to model conditional distributions $Y|X$ via conditional generative models, i.e., $y = G(x)_{*}P_Z$ of some latent distribution $P_Z$. First empirical convergence rates of so-called sieve MLE estimators are provided (Theorem 1), showing that as paired data tends to infinity the approximation of the measure tends to zero in probability.  Then also the Wasserstein distance between the pushforward distribution of the MLE estimate and the true generated distribution can be controlled via a pertubation strategy that depends on the data dimension and smoothness. Then with some ansatz function class spaces for the true generating process, a convergence theorem for sieve MLE with generating functions in some sparse NN class is presented.

Finally, the sieve MLE is validated on some toy examples which is learned via a VAE and on the MNIST dataset.

**Strengths:**

The paper tackles a very interesting topic. There has not been much theory of conditional generative models, in particular not on their statistical rates with respect to the true smoothness and intrinsic dimension.

Theorem 1 presents a very nice approach on how to tackle these problems, which allows for splitting these problems up into the covering number and the accuracy of the approximation. This gives a nice guideline: Pick a function class that has a good covering scaling and good approximation capabilities. In particular the noise injection strategy seems very neat in corollary 2, if the data distribution is close to a singular one. Generally, the writing and the maths is very rigoruous, with clearly defined objects up to a few exceptions (see weaknesses). It is also nice that the qualitiative behaviour of the theorems is verified in understandable toy examples.

**Weaknesses:**

First, let me say, that I come from a background in conditional generative models, not so much in statistical theory. Therefore I had difficulties following some claims in the paper.

1) First, when I think of where conditional generative models are actually used, I am confused about the whole modelling assumption in section 2.1. Usually the latent space Z should take care of the uncertainty in Y, so that T(x,Z) models this perfectly. of course for trained models there will still be a mismatch, but treating this as an independent Gaussian seems very unrealistic. Can you elaborate this assumption maybe taking the example of inverse problems?

Can you maybe find for lets say a normalizing flow always a similar regular normalizing flow with a small gaussian noise added to it such that it becomes the "real" generative model? This would increase my confidence in this assumption.

2) It would be good to have a running example, something super simple like taking the pushforward of a Gaussian under a linear operator (with non trivial rank) to explain the rates and corollaries.

3) I am worried about the scaling in corollary 1 for the $\sigma^{*}$, basically this means that if the noise level tends to zero, that the approximation accuracy has to increase by a lot. This might be due to the choice of the Hellinger distance. Why not try to control the Wasserstein distance in equation 7 in the first space? Since $W_1(p, p \star \mathcal{N}(0,\sigma))$ are close for small $\sigma$ this might alleviate this problem.

4) I feel like section 2.4 is not rigorous in the sense that it is very unclear if the article looks at joint measures of $(X,Y)$ or conditional measures $Y|X=x$. Looking at the proofs I think it holds for the condiitonals and arbitrary $X=x$? ´

5) For the experiments part, it would be very enlightening to consider manifolds with a known dimension (and maybe make them conditional) to validate how the intrinsic dimension comes into play [1]. Further, for the MNIST example, one does not need to take the MNIST dataset as the ground truth, but rather one can take a pretrained generator as the ground truth and then validate how smoothness and approximation error impact the Wasserstein/FID/... between the pretrained generator and a learned model. This would be much more in spirit with validating the bounds, since the point of this paper is not to come up with a practical algorithm I presume.

6) Can the in probability bounds be strengethened to expectation bounds over x between the conditional measures under stronger assumptions, see also [4].Conditional Wasserstein Distances with Applications in Bayesian OT Flow Matching

7) Also, one algorithm where the composition approach might come into play is flow matching/diffusion with odes [2]. Here, if one takes a simple Euler scheme to sample one indeed has such a simple compositional structure. I would find it very interesting to briefly state the relation to those, as they are bit more SOTA and practical interest than VAEs at the moment.

8) Can you make a practical example where the smoothness disparity is obvious?

9) This paper is missing some citations on generative models and manifolds [1,3], and the relation between conditional measures and joint measures [4] and also in what sense conditional generative models optimize the MLE estimator (somewhat clear for NFs and VAEs, less clear for diffusion/...). Here I am thinking along the lines of [5]

Overall, I am mainly missing practical takeaways and a running example where I can see that the assumptions made are indeed practical. Indeed, also the modelling assumptions remain unclear, and would greatly benefit from more explanations. I feel like the whole paper can be a bit streamlined to make it more digestible for the learning community as opposed to the statistics community, but overall I appreciate the deep theoretical analysis if my points can be addressed.

[1] Diffusion Models Encode the Intrinsic Dimension of Data Manifolds, ICML

[2] Flow Matching for Generative Modeling, ICLR

[3] Score-Based Generative Models Detect Manifolds, NeurIPS

[4] Conditional Wasserstein Distances with Applications in Bayesian OT Flow Matching, arXiv

[5] Maximum Likelihood Training of Score-Based Diffusion Models, NeurIPS

**Questions:**

see the weaknesses, they are a bit intertwined.

POST REBUTTAL: Concerns have been mostly addressed. raising score from 6 to 8.

---

> ### Author Response · Authors · 2024-11-22
>
> We sincerely thank the reviewer for their thoughtful comments and constructive feedback. We greatly appreciate their recognition of the novelty and rigor of our approach.
>
> Below, we provide detailed responses to each of the points raised.
>
> 1. We assume the data-generating process $ Y = G_*(Z, X) + \varepsilon $, where $ G_* $ maps latent variables $ Z $ and covariates $ X $ to the manifold, while $ \varepsilon $ captures noise. The objective is to recover $ G_* $ to generate data that lies exactly on the manifold. However, real-world data is inherently noisy and does not perfectly sit on the manifold, making assumptions that embed noise into the latent space overly restrictive.
>
>     Ignoring $ \varepsilon $ and embedding it into the generator would blur the distinction between the signal and noise, undermining the deconvolution objective. For instance, consider data near a 2-sphere in 3D, where a binary covariate $ X $ indicates the upper ($ X = 1 $) or lower ($ X = 0 $) hemisphere. While $ G_* $ should map $ Z $ and $ X $ to points on the 2-sphere, observed responses deviate due to noise. The key challenge is disentangling $ \varepsilon $ while ensuring $ G_* $ faithfully captures the manifold structure.
>
> 2.  A simple example is the generator map $ u \mapsto u^{1/(\beta+1)} $, which transforms a 1D uniform random variable into one with a $ \beta $-Hölder smooth density. Here, setting $ \beta_* = \beta $ and $ t_* = 1 $ allows us to reinterpret the results in our framework, offering a clear illustration of the theoretical findings.
>
> 3. When the noise level $ \varepsilon $ approaches zero, likelihood-based approaches face significant challenges due to the singularity of the support space for $ Y $, requiring precise approximations and making optimization difficult. Controlling the $ W_1 $ distance is equally non-trivial, as it involves solving computationally intensive likelihood functions and requires fundamentally different theoretical tools, including optimal transport. We appreciate this insightful point and are actively working on a long-term project to derive convergence rates and minimax bounds for likelihood-based methods under the $ W_1 $ distance.
>
>     This is a complex yet promising direction, and we are also exploring diffusion- and flow-based models that leverage score-based and velocity field techniques to address these challenges.
>
> 4. Our work focuses on conditional models, with all results holding uniformly for all $ x \in \mathcal{X} $. Lemma 3 (Noise Outsourcing) establishes the existence of a generator within the distribution regression framework. Section 2.4 characterizes the class of conditional distributions achieved via the push-forward of the generator $ G_* $, showing that the learnable distribution class is broad and includes many commonly used distribution classes, demonstrating the generality and flexibility of our approach.
>
> 5. In our experiments, we used two conditional manifold examples—the ellipse and two-moon examples—to validate the trends in Corollary 2.
>
>     We agree that exploring intrinsic dimensionality with simple examples and pretrained models for generating MNIST images would enhance clarity and validate theoretical bounds. However, due to time constraints during the rebuttal period, we defer this to future work and look forward to incorporating it into our ongoing research on conditional flow matching. Thank you for the insightful suggestion!
>
> 6. Please refer to the definition of the integrated Wasserstein distance in Section 2.3, just before Theorem 2. This metric is defined as the expectation over the distribution of $ X $. It corresponds to the same metric as Proposition 1.(i) in [4], with the specific case of $ p = 1 $.
>
> 7. Thank you for highlighting these approaches. While score-based methods were briefly mentioned in our introduction, to our knowledge, there are no theoretical results addressing composite structures in flow-matching or diffusion-based models. On a related note, our group is exploring how to integrate composite structures into flow-matching frameworks, particularly investigating the smoothness properties of linear flow vector fields for composite-structured target distributions.
>
> 8. Two practical examples illustrating the smoothness disparity framework are **Heart Rate Variability** and **Air Quality Modeling**: In heart rate variability, the cardiac response conditioned on physical activity exhibits irregular, non-smooth patterns, while physical activity transitions are smoother. Similarly, in air quality modeling, particulate matter (PM) concentrations conditioned on temperature often show sharp spikes (e.g., from pollution or fires), while temperature changes are generally smoother.

---

> > ### Author Response · Authors · 2024-11-22
> >
> > 9. Thank you for sharing these valuable citations on generative models and manifolds. We have incorporated them into the updated manuscript to enhance its depth and relevance. As we understand, VAEs and NFs aim to approximately minimize the true likelihood, while diffusion models optimize the score, i.e., the gradient of the log-likelihood. Regarding [5], it appears the authors use a variational approach to approximate the true score, similar to how variational inference approximates the true likelihood.
> >
> > A key takeaway from our work is that the derived rate is near-optimal while also providing a theoretical explanation for the ability of conditional generative models to learn manifolds and composite structures. This bridges the gap between the vast empirical advancements in the field and a rigorous theoretical foundation.
> >
> > Assumption 1 ensures the existence of a density for both the noisy data and its noiseless counterpart on the manifold. This is a standard and widely accepted assumption in the statistical and machine learning literature. Assumptions 2 and 3 have strong practical motivations. For instance, [H] highlights the existence of lower-dimensional structures in real-world data, offering empirical evidence that supports these assumptions. Specifically, Assumption 2 generalizes commonly observed data structures, such as multiplicative and additive models, as discussed in [I].
> >
> > - [H] The Intrinsic Dimension of Images and Its Impact on Learning (arXiv:2104.08894)
> > - [I] Generalized Additive Models: Some Applications (JASA, 1987)

---

> > > ### Comment · Reviewer_VL1U · 2024-11-23
> > > **Rebuttal**
> > >
> > > Thanks for the rebuttal. regarding 1) I still think that it is not sensible to model it with additive Gaussian noise. Lets say you train a conditional normalizing flow for the inverse problem $X = Y + \varepsilon$, then, in the optimal case, one has that $P_{Y|X=x} = T(x)_{*}P_Z$.  This means that you can write $Y = T(X,Z)$ (i.e. no noise!). If you have an unperfect conditional NF, then I remain unconvinced that the difference $Y - T(X,Z)$ is an additive Gaussian, which would mean that the conditional generator makes errors in a very structured (or rather unstructured) way. I dont think the noise "manifold" argument is really applicable here, since also the conditional NF handles noisy data without an additional $\varepsilon$ term. The uncertainty stems from sampling $Z$.
> > >
> > > I would think that the argument I outlined would still work "Can you maybe find for lets say a normalizing flow always a similar regular normalizing flow with a small gaussian noise added to it such that it becomes the "real" generative model? This would increase my confidence in this assumption."
> > >
> > > I.e., take a conditional NF, which makes a non Gaussian error, and then approximate it with a one so that the residual becomes Gaussian.
> > >
> > > Other than that, I will read over the new manuscript, but my other points seem to be well-addressed.

---

> > > > ### Author Response · Authors · 2024-11-25
> > > >
> > > > Thank you for bringing this up. We approach the problem from the perspective of manifold learning, highlighting key references such as [Genovese1], [Genovese2], and [Fan], which address related challenges involving additive noise in the manifold context. However, we emphasize that the noiseless case, where $\epsilon = 0$, constitutes a fundamentally different problem, which lies outside the scope of this work.
> > > >
> > > > Our generative model has two main components: learning the data on the manifold and modeling the empirical noise. A straightforward analogy is linear regression, where one estimate $\widehat{\beta}$ and subsequently recover $\widehat{\epsilon}$ as residuals. Similarly, our generative model aims to capture the underlying manifold structure while accounting for noise in the observations.
> > > >
> > > > From a training standpoint, an ideal generative model (e.g., VAE) is expected to generate data that lies exactly on the manifold, despite the observed responses being perturbed by noise $\epsilon$. However, embedding noise directly into the latent structure would make theoretical analysis of manifold learning infeasible. Specifically, in the generative model $Y = G_*(X, Z)$, the support of the manifold is lost because both $Y$ and $G_*(\mathcal{X}, \mathcal{Z})$ are in $\mathbb{R}^D$, making it impossible to disentangle and analyze the manifold structure.
> > > >
> > > > - [Fan] On the optimal rates of convergence for nonparametric deconvolution problems. AoS 1991 (1257-1272)
> > > > - [Genovese1] Minimax manifold estimation. JMLR 2012 (1263-1291)
> > > > - [Genovese1] Manifold estimation and singular deconvolution under Hausdorff loss. AoS 2012 (941-963)

---

> > > > > ### Comment · Reviewer_VL1U · 2024-11-25
> > > > > **response**
> > > > >
> > > > > Ok, I think I understand, but I feel this assumption is not perfectly realistic, but I understand that this would make the theoretical analysis much harder. Still I believe, one might get around it with using universality results of conditional generators and taking a second generative model approximating the difference between additive gaussian noise and the "first" generator.

---

> > > > > > ### Author Response · Authors · 2024-11-25
> > > > > >
> > > > > > Thank you for your understanding. To the best of our knowledge, this idea of a two-hierarchical generative model has not been extensively explored in the theoretical analysis of conditional generative models. However, it presents an intriguing and promising direction for future research.

---

> ### Author Response · Authors · 2024-11-27
> **A Gentle Reminder**
>
> Dear Reviewer VL1U,
>
> As we approach the final stages of revising our rebuttal PDF, we wanted to gently remind you of our responses to your valuable feedback.
>
> We have carefully addressed all your comments and questions and hope that our efforts **meet your expectations**. If you have any remaining concerns, we would be happy to **discuss them further** during the extended discussion period. Otherwise, if our updates satisfactorily address your feedback, we kindly invite you to consider **raising your score**.
>
> We truly appreciate the time and effort you have dedicated to reviewing our paper, especially given its depth and complexity. Regardless of the final outcome, we are deeply grateful for your thoughtful insights, which have significantly contributed to improving our work.
>
> Thank you again!
>
> Sincerely,
> The Authors

---

> > ### Comment · Reviewer_VL1U · 2024-11-27
> > **rebuttal**
> >
> > thanks for the reminder. I think this paper is in a better shape, raising my score to 8!

---

> > > ### Author Response · Authors · 2024-12-02
> > >
> > > We are glad to hear that we have addressed your concerns and met your expectations. Thank you once again for your valuable contribution in helping us improve the manuscript.
> > >
> > > Your feedback has been very helpful in refining our work, and we truly appreciate the time and effort you have invested.

---

### Official Review · Reviewer_qXt2 · 2024-11-03

**Soundness:** 2
**Presentation:** 2
**Contribution:** 2
**Rating:** 5
**Confidence:** 2

**Summary:**

The authors describe a method for estimating high dimensional conditional distributions that concentrate around a manifold. Such distributions appear in common settings such as conditional generative modeling. The authors consider the convergence of an estimator that improves based on the number of data points to the true distribution in the Hellinger distance sense. The authors then consider applying this estimator to some synthetic and real datasets to understand its empirical performance. The analysis considers two types of neural networks: a sparse neural network and fully connected one that parameterize the push forward of the latent variable to the conditional outcome. The analysis is heavily dependent on empirical process theory to establish different properties of the architectures.

**Strengths:**

The proposed analysis is helpful in understanding why generative models work as they do. Additionally the proposed error analysis is useful for anticipating the number of data points needed to achieve a specific targeted error with respect to the Wasserstein or Hellinger distances.

The empirical results support the main claims of the work to some extent.

**Weaknesses:**

Some of the theoretical results seem hard to verify in practice. It would be interesting for some contrived examples if the authors can show the convergence of the distances as a function of the number of points.

The theorems contain some constants that do not have a clear way of being estimated making some of the errors hard to use in practice.

Note that the paper does not appear to be in the right ICLR format since the margins are considerably smaller.

**Questions:**

How easy are some of the conditions to verify for the convergence theorems to hold?

How do you interpret the constants in the various theoretical statements? For example, some are dependent on the domain or other parameters, in what sense do they grow for perturbations of their dependencies? Can these be estimated in practice?

---

> ### Author Response · Authors · 2024-11-22
>
> We sincerely thank the reviewer for for their thoughtful and detailed feedback and recognizing the relevance of our framework. We appreciate their acknowledgment of the theoretical results, error analysis, and empirical findings.
>
> > Some of the theoretical results seem hard to verify in practice. It would be interesting for some contrived examples if the authors can show the convergence of the distances as a function of the number of points.
>
> We may consider a simple representative example to better understand the obtained rates:
>
> Take the function with density $ f(y) = (\beta + 1)y^{\beta} $, supported on $[0,1]$. This density is Hölder smooth with parameter $\beta$, and the ambient dimension is $1$. When $\alpha = 0$, the rates outlined in Corollary 1 simplify to $ n^{-\beta/(2\beta+1)} $. This corresponds to the minimax optimal rate for estimating $\beta-$Hölder  density function.
>
>
>
> > Note that the paper does not appear to be in the right ICLR format since the margins are considerably smaller.
>
> We apologize for the formatting issue in the original submission and have made the necessary corrections in the revised version. For a detailed overview of the changes, please refer to the global rebuttal, which outlines the specific revisions and adjustments made.
>
> > How easy are some of the conditions to verify for the convergence theorems to hold?
>
> - Assumption 1 ensures the existence of a density for both the noisy data and its noiseless counterpart on the manifold. This is a widely accepted and foundational assumption in statistical and machine learning literature.
> - Assumptions 2 and 3 can be practically motivated in various ways. For instance, [H] explores the existence of lower-dimensional structures in data, providing empirical evidence for such assumptions. Assumption 2, in particular, generalizes common data structures like multiplicative and additive models [I].
>
> For specific problems, certain aspects of these assumptions can be known or approximated. For example, in the MNIST dataset, the ambient dimension $D = 784$ is readily available. However, determining intrinsic characteristics, such as the intrinsic dimension $d$, remains a difficult task.
>
> To the best of our knowledge, there is no existing result that provides a framework for consistent testing of these assumptions while simultaneously achieving efficient estimation. This dual goal represents a challenging open problem that warrants further research.
>
> > The theorems contain some constants that do not have a clear way of being estimated making some of the errors hard to use in practice.
>
> > How do you interpret the constants in the various theoretical statements? For example, some are dependent on the domain or other parameters, in what sense do they grow for perturbations of their dependencies? Can these be estimated in practice?
>
> When analyzing the large sample properties of estimators, the explicit dependence on constants is often deliberately hidden. This approach allows us to focus on the obtained rates without being distracted by the cumbersome and sometimes "ugly" expressions of these constants.
>
> The constants in our theoretical statements depend on fixed model parameters (e.g., $\beta_*$, $t_*$, or $\sigma_{\max}$), which do not grow with sample size. While they can, in principle, be traced through the proofs, estimating them in practice is challenging due to unknown parameters. In specific cases, such as when the data-generating process is known (e.g., $f(y) = 6y^5$), these constants can be directly identified. However, adaptively estimating these parameters with theoretical guarantees in a non-parametric setting remains a complex open problem. In practical applications, partial knowledge—such as ambient dimensions (e.g., $D = 784$ in MNIST)—can provide some guidance, but estimating intrinsic dimensions like $d$ is notably harder and has been the focus of significant research such as [H].
>
> - [H] The Intrinsic Dimension of Images and Its Impact on Learning (arXiv:2104.08894)
> - [I] Generalized Additive Models: Some Applications (JASA, 1987)

---

> > ### Author Response · Authors · 2024-11-27
> > **A Gentle Reminder**
> >
> > Dear Reviewer qXt2,
> >
> > As we approach the final stages of revising our rebuttal PDF, we wanted to gently remind you of our responses to your valuable feedback.
> >
> > We have carefully addressed all your comments and questions and hope that our efforts **meet your expectations**. If you have any remaining concerns, we would be happy to **discuss them further** during the extended discussion period. Otherwise, if our updates satisfactorily address your feedback, we kindly invite you to consider **raising your score**.
> >
> > We truly appreciate the time and effort you have dedicated to reviewing our paper, especially given its depth and complexity. Regardless of the final outcome, we are deeply grateful for your thoughtful insights, which have significantly contributed to improving our work.
> >
> > Thank you again!
> >
> > Sincerely,
> > The Authors

---

> > > ### Author Response · Authors · 2024-12-02
> > > **Another Gentle Reminder**
> > >
> > > Dear Reviewer qXt2,
> > >
> > > As the discussion period approaches its conclusion, we would greatly appreciate your feedback on our rebuttal.
> > >
> > > In our response, we have made every effort to address all of your concerns and additional questions.
> > >
> > > If you feel that your concerns have been adequately addressed, we kindly invite you to consider a score adjustment. If there are still any unresolved issues, we would be more than happy to use the remaining time to provide further clarifications.
> > >
> > > Thank you for your time and consideration.
> > >
> > > Best regards,
> > > The Authors

---

### Official Review · Reviewer_Qywh · 2024-11-04

**Soundness:** 3
**Presentation:** 1
**Contribution:** 3
**Rating:** 5
**Confidence:** 4

**Summary:**

This paper studies using neural network generative models for distribution regression. Statistical rates are established, and a significant effort is devoted to resolving the curse of dimensionality issue. Numerical results are provided to support the theory.

**Strengths:**

The theoretical results appear to be correct and sound. Moreover, there are detailed analyses for different setups and matching lower bound.

There are numerical results to support the performance of neural network generative models for distribution regression.

**Weaknesses:**

The paper is not very easy to follow in places, mostly due to the technical presentations and pure statistical discussions. I would suggest authors to prepare a preliminary section before mixing background introduction with main results in current Section 2.

It is relatively difficult for me to judge the technical novelty of the paper. On the one hand, the distribution regression problem and low-dimensional generative neural networks seem to pose technical difficulties. On the other hand, the results as well as most of the assumptions are replication of many existing works, such as Schmidt-Hieber, 2020 and Tang and Yang, 2023.

Practical impact of this work may be limited, due to many technical assumptions.

Numerical results loosely connect to the theoretical results.

The paper is not prepared in the standard ICLR style --- the left and right margins are altered significantly. Under the standard style template, this submission will go beyond the page limit.

**Questions:**

Do numerical results support the rates proved in theory? Maybe we can take logarithm on the sample size and inspect the slope of the error.

Assumption 3 excludes interior points. Does this assumption hold (approximately) in practical applications?

---

> ### Author Response · Authors · 2024-11-23
>
> We sincerely thank the reviewer for their feedback. We greatly appreciate their recognition of our detailed analyses across various setups.
>
> > The paper is not very easy to follow... . It is relatively difficult for me ...
>
> Thank you for your suggestion. The focus of this paper is on characterizing the statistical foundations of conditional deep generative models by rigorously studying their statistical properties. We aim to understand why these models perform so well by providing a unified framework that extends and generalizes several previous studies, such as the work by Chae et al. (2023) which focuses on the case where $|\mathcal{X}| = 1$.
>
> Our approach differs from existing works like Schmidt-Hieber (2020), which uses sparse neural networks for regression in a non-generative setting. In contrast, we address a broader problem and incorporate additional mechanisms like the manifold hypothesis to handle the curse of dimensionality. Our framework provides rigorous guarantees for both sparse and fully connected networks.
>
> Additionally, Tang and Yang (2023) rely on adversarial or GAN-based methods with strict assumptions that the data lies exactly on the manifold and has a lower-bounded density. Our approach is more flexible, allowing noisy data close to the manifold without requiring a lower-bounded density, and offers practical advantages over adversarial-based methods.
>
> > Practical impact ... assumptions.
>
> Please allow me to illustrate the technical assumptions and their practical implications.
>
> - Assumption 1 ensures the existence of densities for both the noisy data and its underlying noiseless counterpart on the manifold, which is a foundational and widely accepted assumption in statistical theory.
> - Assumption 2 introduces structural constraints to overcome the curse of dimensionality. This assumption leverages inherent data structures, such as additive models, which are widely used in practice [E].
> - Assumption 3 employs the manifold hypothesis to further reduce the curse of dimensionality, with applications in image data, which naturally reside on low-dimensional manifolds within high-dimensional spaces [C]. The concept of reach, as the inverse of the manifold's condition number, is crucial here, and while no consistent manifold estimation methods exist for manifolds with zero reach, this challenge is well understood in manifold learning literature [F,G].
>
> > Numerical results loosely connect to the theoretical results.
>
> Thank you for your comment. The primary focus of this work is theoretical, with the numerical examples serving to illustrate and validate the proposed methodology in practical settings. We provide three types of examples to demonstrate different aspects: the FD examples validate the methodology in full-dimensional settings, the toy manifold examples confirm the trends outlined in Corollary 2, and the MNIST example highlights a scenario where the intrinsic dimension is much smaller than the ambient data dimension ($D = 784$). These examples are designed to offer statistical insights and confirm the broader applicability of our theoretical results.
>
> > The paper is not prepared in the standard ICLR style ...
>
> We apologize for the formatting issue in the original submission and have corrected it in the revised version. Please refer to the global rebuttal for a detailed overview of the revisions made.
>
> > Do numerical results support the rates proved in theory? Maybe we can take logarithm on the sample size and inspect the slope of the error.
>
> Following your suggestion, we extended our analysis to examine how the $W_1$ distance varies with sample size, while keeping the noise level fixed at $\sigma_* = 0.01$. Below is a summary table showing the median empirical Wasserstein distances for different sample sizes. The experimental setup remains consistent with the manifold case described in the manuscript.
>
> |Sample Size|Two Moon (with $\sigma_* = 0.01$)|Ellipse (with $\sigma_* = 0.01$)|
> |-|-|-|
> | 4000 | 0.251| 0.295|
> | 6000 | 0.232| 0.285|
> | 7000 | 0.216| 0.271|
> | 8000 | 0.214| 0.253|
> | 9000 | 0.212| 0.259|
> | 10000| 0.196| 0.251|
>
> While extracting exact rates can be challenging, the results in the table validate the large-sample properties for manifolds. These empirical findings align well with the theoretical expectations, further confirming the consistency and convergence trends of our framework.
>
>
> > Does Assumption 3, which excludes interior points, hold (approximately) in practical applications?
>
> Yes, Assumption 3 holds in practical applications. It ensures that the closure of the manifold is well-defined, which is a common requirement in manifold learning literature [D]. This assumption is crucial for maintaining the low-dimensional structure of the data. If the manifold were space-filling in $\mathbb{R}^D$, the data would lose its low-dimensional properties. In practice, it implies that data points lie close to a manifold with a well-defined structure, as discussed in [C].

---

> > ### Author Response · Authors · 2024-11-23
> >
> > - [C] The Intrinsic Dimension of Images and Its Impact on Learning (arXiv:2104.08894)
> > - [D] Estimating the Reach of a Manifold via Its Convexity Defect Function (arXiv:2001.08006)
> > - [E] Generalized Additive Models: Some Applications (JASA, 1987)
> > - [F] Vector Diffusion Maps and the Connection Laplacian (arXiv: 1102.0075)
> > - [G] Estimating the Reach of a Manifold (arXiv: 1705.04565)

---

> > > ### Author Response · Authors · 2024-11-27
> > > **A Gentle Reminder**
> > >
> > > Dear Reviewer Qywh,
> > >
> > > As we approach the final stages of revising our rebuttal PDF, we wanted to gently remind you of our responses to your valuable feedback.
> > >
> > > We have carefully addressed all your comments and questions and hope that our efforts **meet your expectations**. If you have any remaining concerns, we would be happy to **discuss them further** during the extended discussion period. Otherwise, if our updates satisfactorily address your feedback, we kindly invite you to consider **raising your score**.
> > >
> > > We truly appreciate the time and effort you have dedicated to reviewing our paper, especially given its depth and complexity. Regardless of the final outcome, we are deeply grateful for your thoughtful insights, which have significantly contributed to improving our work.
> > >
> > > Thank you again!
> > >
> > > Sincerely,
> > > The Authors

---

> > > > ### Author Response · Authors · 2024-12-02
> > > > **Another Gentle Reminder**
> > > >
> > > > Dear Reviewer Qywh,
> > > >
> > > > As the discussion period approaches its conclusion, we would greatly appreciate your feedback on our rebuttal.
> > > >
> > > > In our response, we have made every effort to address all of your concerns and additional questions.
> > > >
> > > > If you feel that your concerns have been adequately addressed, we kindly invite you to consider a score adjustment. If there are still any unresolved issues, we would be more than happy to use the remaining time to provide further clarifications.
> > > >
> > > > Thank you for your time and consideration.
> > > >
> > > > Best regards,
> > > > The Authors

---

> > > > > ### Comment · Reviewer_Qywh · 2024-12-02
> > > > >
> > > > > Sorry for a late reply. I appreciate authors' responses to my questions and concerns. I have raised my score to a 5.
> > > > >
> > > > > Some factors that prevent a direct positive rating are
> > > > >
> > > > > 1. The theoretical study does not have much practical implications. Although the assumptions can be understood as abstractions of practical scenarios, the results do not provide immediate algorithm innovation or network architecture design for applications.
> > > > >
> > > > > 2. The added numerical results validate the trend of reduced errors as the sample size increases. I believe this is to be expected and relatively independent of the sample complexity developed in the paper.
> > > > >
> > > > > To sum up, I believe this paper provides solid statistical analysis to an interesting question. Yet its impact to deep generative models as well as deep learning is limited.

---

> ### Author Response · Authors · 2024-12-03
>
> Dear Reviewer Qywh,
>
> Thank you for your thoughtful comments and for raising your score. We sincerely appreciate your acknowledgment of our theoretical contributions and statistical analysis.
>
> While our study focuses on foundational theory rather than immediate practical applications, we hope it serves as a stepping stone for future research to bridge the gap between theory and empirical advancements. We are confident that our work can inspire further exploration, leading to algorithmic innovations and architectural designs.
>
> Sincerely,
>
> The Authors

---

### Official Review · Reviewer_uhNp · 2024-11-04

**Soundness:** 4
**Presentation:** 4
**Contribution:** 3
**Rating:** 6
**Confidence:** 4

**Summary:**

A statistical analysis of conditional deep generative models using a likelihood approach (specifically a sieve MLE) is proposed and analyzed. A convergence rate for the estimator is established using neural network approximation theory for the (conditional) generator. Numerical examples are provided showing the efficacy of the method.

**Strengths:**

1. Very well written paper
2. Method is clearly explained and proof are followable

**Weaknesses:**

1. The method seems to be extending the techniques of previous work and filling in some details with approximation.
2. Question on the applicability to multiple chart manifolds.
3. Numerical examples are very simple.

**Questions:**

1. In regards to the discussion in 2.4.3 and Appendix C with the multiple manifold case: If a partition of unity is used to define local transport maps, we require that the local distributions on each chart need to have density lower bounded. However, a partition of unity will have its weights vanish to 0 on the boundary, so the induced local distribution would have density that vanishes to 0 on the boundary. How can you apply optimal transport regularity theory in that case, when your target has density that is not lower bounded?

---

> ### Author Response · Authors · 2024-11-22
>
> We thank the reviewer for their valuable feedback and sincerely appreciate their positive remarks on the clarity of the presentation and the rigor of the proofs in our manuscript.
>
>
> > 1. The method seems to be extending the techniques of previous work and filling in some details with approximation.
>
> Our work aims to address the gap in understanding why conditional deep generative models perform well by examining the problem within a statistical framework. Beyond bridging this gap, our cohesive framework not only provides key insights but also generalizes several existing works. A notable special case of our results is Chae et al. (2023), where the model complexity is significantly smaller, as they only consider the case when $|\mathcal{X}| = 1$. In contrast, our work allows for infinitely many co-variates simultaneously. Another special case (derived from Theorem 3) is Li et al. (2022), which examines sequential smoothness structures but in a non-generative model framework.
>
> From a technical standpoint, we have introduced several novel tools in our proofs, which are elaborated upon in the appendix, while also leveraging established techniques with proper citations to the best of our knowledge. Notable contributions include the integration of two distinct classes of neural networks—sparse and fully connected—into a unified framework, the derivation of a general convergence result for distribution regression as stated in Theorem 1, and the development of a novel approximation result, which is rigorously established in Theorem 5.
>
> > 2. Question on the applicability to multiple chart manifolds.
>
> > 1. In regards to the discussion in 2.4.3 and Appendix C with the multiple manifold case: If a partition of unity is used to define local transport maps, we require that the local distributions on each chart need to have density lower bounded. However, a partition of unity will have its weights vanish to 0 on the boundary, so the induced local distribution would have density that vanishes to 0 on the boundary. How can you apply optimal transport regularity theory in that case, when your target has density that is not lower bounded?
>
> Thank you for pointing this out. We have made a small modification in the proof to address the issue (please see the updated manuscript). This correction ensures that the transport map properly handles densities that are both upper and lower bounded, maintaining the desired properties throughout the mapping process.
>
> > 3. Numerical examples are very simple.
>
> The focus of this work is primarily theoretical in nature. The examples provided are intended to validate the proposed methodology and offer statistical insights in different settings. We utilize the (conditional) VAE architecture to demonstrate the applicability of our theoretical findings but do not propose any new algorithms. The developed theory is equally applicable to other likelihood-based methods, such as Normalizing Flows.
>
> On the algorithmic and computational side, there are existing works, such as [A] and [B], that incorporate more advanced examples; however, these primarily focus on computational performance and offer limited statistical insights.
>
>  - [A] Conditional Variational Autoencoder for Neural Machine Translation (arXiv:1812.04405)
>  - [B] Learning Likelihoods with Conditional Normalizing Flows (arXiv:1912.00042)

---

> ### Comment · Reviewer_uhNp · 2024-11-25
>
> Thank you for your response and updating the manuscript. I had a question about Appendix D of the revision. While the formula for the density $q$ in line 945 follows from the partition of unity (as it involves a sum of scalar valued quantities, i.e. the probability density), I am confused about the for the formula for $G$ in line 964. Correct me if I am wrong, but the co-domain of $\varphi_k \circ g_k$ should be the patch of the manifold $U_k \subset \mathcal{Y}$. Reducing to the case of $K=2$, the formula is (where $h_k = \varphi_k \circ g_k$)
>
> $$G(z,x) = \pi_1 \tau_1(h_1(z, x)) h_1(z, x) + \pi_2 \tau_2(h_2(z, x)) h_2(z, x)$$
>
> $h_k(z, x)$ is an element of $U_k$, and the quantity $\pi_k \tau_k(h_k(z, x))$ is a scalar, so the RHS involves scaling elements of $U_1$ and $U_2$ and adding them together. I think the adding together part doesn't matter since the domains $\mathcal{V}_k$ are disjoint, but I do not know how to scale an element of a manifold. In other words, I am confused about how to interpret the value $\tau_1(h_1(z, x)) h_1(z, x)$ if $\tau_1(h_1(z, x)) \neq 1$. This makes me think that the formula in line 964 is not correct.

---

> > ### Author Response · Authors · 2024-11-26
> > **Response: correction in the manifold proof**
> >
> > Thank you for catching this mistake in the proof. Patching multiple transport maps while ensuring regularity guarantees is indeed a challenging task. Therefore, we adhere to the original strategy of defining transport maps for the mixture distribution. Considering your comment about the boundary densities not being lower bounded, we are addressing this issue by relying on weaker results (please refer to the revised manuscript). We hope this resolves your concern, and we are happy to address any further questions or concerns you may have.

---

> > > ### Author Response · Authors · 2024-11-27
> > > **A Gentle Reminder**
> > >
> > > Dear Reviewer uhNp,
> > >
> > > As we approach the final stages of revising our rebuttal PDF, we wanted to gently remind you of our responses to your valuable feedback.
> > >
> > > We have carefully addressed all your comments and questions and hope that our efforts **meet your expectations**. If you have any remaining concerns, we would be happy to **discuss them further** during the extended discussion period. Otherwise, if our updates satisfactorily address your feedback, we kindly invite you to consider **raising your score**.
> > >
> > > We truly appreciate the time and effort you have dedicated to reviewing our paper, especially given its depth and complexity. Regardless of the final outcome, we are deeply grateful for your thoughtful insights, which have significantly contributed to improving our work.
> > >
> > > Thank you again!
> > >
> > > Sincerely,
> > > The Authors

---

> > > > ### Author Response · Authors · 2024-12-02
> > > > **Another Gentle Reminder**
> > > >
> > > > Dear Reviewer uhNp
> > > >
> > > > As the discussion period approaches its conclusion, we would greatly appreciate your feedback on our rebuttal.
> > > >
> > > > In our response, we have made every effort to address all of your concerns and additional questions.
> > > >
> > > > If you feel that your concerns have been adequately addressed, we kindly invite you to consider a score adjustment. If there are still any unresolved issues, we would be more than happy to use the remaining time to provide further clarifications.
> > > >
> > > > Thank you for your time and consideration.
> > > >
> > > > Best regards,
> > > > The Authors

---

### Author Response · Authors · 2024-11-22

We would like to sincerely thank all four reviewers for their time and thoughtful feedback. Their insightful comments have significantly improved the quality of our revised manuscript. We also appreciate their recognition of the strength and clarity of our theoretical results, as well as the accessibility of the accompanying proofs.

First and foremost, we apologize for the formatting issue in the original submission. This oversight stemmed from the use of `usepackage\{fullpage}`. To address this while adhering to the page limit restrictions, we have made the following updates to the manuscript:

- Moved the MNIST dataset numerical results to the appendix.
- Adjusted formatting for readability and compactness, particularly in lines `456–460` and `489–493`.
- Verified that the final version of the manuscript, including the author names, is well within the 10-page limit.
- Corrected the proof in the multiple-chart manifold case (lines `893-917`) to ensure the transported density remains lower bounded.

Additionally, we conducted further numerical studies to address some reviewers' comments, specifically analyzing how the $W_1$ distance varies with sample size, while keeping the noise level fixed at $\sigma_* = 0.01$. The experimental setup is consistent with the manifold case described in the manuscript. Below is a summary of the empirical results:

| Sample Size | Two Moon (with $\sigma_* = 0.01$) | Ellipse (with $\sigma_* = 0.01$) |
|-------------|----------------------------------|--------------------------------|
| 4000        | 0.251                            | 0.295                          |
| 6000        | 0.232                            | 0.285                          |
| 7000        | 0.216                            | 0.271                          |
| 8000        | 0.214                            | 0.253                          |
| 9000        | 0.212                            | 0.259                          |
| 10000       | 0.196                            | 0.251                          |

Although exact rates are difficult to extract, these results validate the large-sample properties for manifolds, aligning well with our theoretical expectations. This further supports the consistency and convergence trends of our framework.

---

### Meta-Review · Area_Chair_xoUt · 2024-12-23

**Metareview:**

This paper presents a theoretical analysis of conditional deep generative models, focusing on estimating high-dimensional conditional distributions that concentrate around manifolds. The authors provide statistical guarantees and convergence rates for different network architectures and present empirical results to support their theoretical claims.

The main strength of the paper lies in its attempt to provide a theoretical foundation for conditional generative models, offering convergence rates and error bounds for different network architectures. However, the paper suffers from several weaknesses. The practical implications and immediate impact on deep learning applications are limited. Many of the assumptions and constants in the theoretical framework are difficult to verify or estimate in practice. Moreover, the theoretical contribution appears to primarily combine existing work without offering significant new insights.

The decision to reject is primarily based on the limited novelty and practical impact of the work. While the paper provides a theoretical analysis, it does not offer substantial new insights or immediate practical applications in deep learning. The theoretical results largely align with expected outcomes and do not demonstrate significant deviations from existing work. Additionally, the initial submission's violation of formatting guidelines raises concerns about adherence to conference standards.

**Additional Comments On Reviewer Discussion:**

During the rebuttal period, reviewers expressed concerns about the practical verifiability of theoretical assumptions, interpretation of constants, and limited practical implications. The authors attempted to address these issues by providing additional context, conducting extra numerical experiments, and clarifying connections to related work.

Although two reviewers raised their scores after the rebuttal, acknowledging the authors' efforts to clarify their work, the core issues of limited novelty and practical impact remained unresolved. Despite the authors' thorough responses and attempts to link their work to practical applications, the fundamental concerns persisted. These factors, combined with the initial formatting violation, ultimately led to the decision to reject the paper.

---

### Decision · Program_Chairs · 2025-01-22

Reject